# Starvation Levels Affect Behaviors of Wild-Caught and Laboratory-Reared Oil Palm Pollinator Weevil, *Elaeidobius kamerunicus* (Coleoptera: Curculionidae)

**DOI:** 10.3390/insects13100940

**Published:** 2022-10-17

**Authors:** Nurul Fatihah Abd Latip, Idris Abd Ghani, Izfa Riza Hazmi, Dzulhelmi Muhammad Nasir

**Affiliations:** 1Faculty of Plantation and Agrotechnology, Universiti Teknologi MARA, Perlis Branch, Arau Campus, Perlis 02600, Malaysia; 2Centre for Insect Systematics, Faculty of Science and Technology, Universiti Kebangsaan Malaysia, Bangi 43600, Malaysia; idrisyatie@yahoo.com.my (I.A.G.); izfahazmi@ukm.edu.my (I.R.H.); 3Crop Protection & Bio-Solutions, FGV R&D Sdn Bhd, Tun Razak Agricultural Research Centre, Jerantut 27000, Malaysia; dzulhelmi.mn@fgvholdings.com

**Keywords:** Coleoptera, Curculionidae, diurnal behavior, *Elaeidobius kamerunicus*, oil palm pollinator, weevil

## Abstract

**Simple Summary:**

*Elaeidobius kamerunicus* Faust (Coleoptera: Curculionidae) was introduced into Malaysia from Cameroon in 1981 to increase oil palm pollination. Fruit set development and fruit production significantly increased after the introduction, until a reported decline that began in the late 1980s. Several factors may have contributed to such a decline, including the weevil’s behavior. In this study, we evaluated the diurnal behavior of *E. kamerunicus* on different starvation level, sexes and sources of *E. kamerunicus* (wild-caught and lab-reared) through two hours of direct observation. The frequency and the time spent for each behavior, namely the flying (F), moving (M), feeding (E), resting (R), grooming (G) and mating (C) on oil palm flowers were recorded and evaluated. There were significant differences in flying, resting and grooming behavior among starvation levels. A similar result was demonstrated by sources of *E. kamerunicus* for resting, feeding and mating behavior. Wild-caught *E. kamerunicus* had a bigger size compared to laboratory-reared *E. kamerunicus* due to environmental and food source factors, which were limited and less variable for the reared type. Overall, our findings indicate that starvation level, sexes and sources of *E. kamerunicus* affect the diurnal behavior of *E. kamerunicus.*

**Abstract:**

The oil palm pollinating weevil, *Elaeidobius kamerunicus* Faust (Coleoptera: Curculionidae) was first introduced into Malaysia from Cameroon as the main oil palm pollinator in 1981. Since then, oil palm pollination has improved and the need for assisted pollination has reduced. However, their pollinating behavior may be influenced by starvation, sexes and sources (wild-caught and lab-reared). A study to determine the effect of starvation level, sexes and sources of *E. kamerunicus* on their diurnal behavior was conducted utilizing two hours of direct observation between 1130 and 1330 h. The frequency and the time spent for several diurnal behaviors on oil palm flowers were recorded and evaluated. Starvation prevented the weevils from conducting the activities because they probably had to focus more on searching for food to gain energy to perform other behavior. Wild-caught and lab-reared *E. kamerunicus* differed in their size, while sexes of *E. kamerunicus* significantly affect the diurnal behavior. However, an in-depth study is needed to determine the relationship between the diurnal behavior of *E. kamerunicus* and oil palm production.

## 1. Introduction

The oil palm, African *Elaeis guineensis* pollinating weevil, *Elaiedobius kamerunicus Faust* (Coleoptera: Curculionidae), was first introduced to Malaysia from Cameroon in 1981 [1]. They were identified as the most efficient insect pollinators for oil palm because they carry more pollen grains than any other insect pollinator species [2] and are well adapted to the wet season [3]. Oil palm is its specific host on which breeding can occur [1]. Since their introduction, the Malaysian oil palm industry has grown significantly, with improvements in oil palm pollination, reducing the need for assisted pollination as well as increasing fruit set development and fruit production [4,5,6,7] and saving tens of millions of pounds on manual pollination led to great success for the oil palm industry [8]. As of December 2019, Malaysia had 5.9 million hectares of oil palm plantations [9]. The percentage of commercial oil palm plantations was 71.7%, while smallholder growers owned the remaining 28.3%. According to Parveez et al. [10], the Malaysian oil palm industry produced 19.86 million tons of crude oil palm (CPO), 17.19 tons per hectare of fresh fruit bunches (FFB), 20.21% of oil extraction rates (OER) and 16.88 million tons of oil palm for export in 2019.

This weevil is attracted to both male and female oil palm flowers due to the anise-like odor of estragole released by both flowers. However, at anthesis, the oil palm male flower released a distinct anise-like odor, which seemed to be stronger compared to the oil palm female flower [11] and it provides a food source and breeding site for *E. kamerunicus* [12]. Furthermore, the oil palm male flower provides nectar and pollen grains [13], while oil palm female flowers just provide nectar, which results in a higher abundance of *E. kamerunicus* on male inflorescence than the female inflorescence. Higher numbers of *E. kamerunicus* gathered on both oil palm flowers during the first day of anthesis, as had been reported previously by Dhileepan [3]. The high abundance of *E. kamerunicus* on the first day of flower anthesis is strongly correlated with the odor of estragole released by both flowers on the first day of anthesis, which decreased as the day of anthesis increased [13].

*Elaeidobius kamerunicus* was known as an insect with diurnal activity. Chiu [14] reported that several studies have been conducted to determine the active time of *E. kamerunicus*. He reported that the *E. kamerunicus* was more active between 1230 and 1430 h while inactive between 0730 and 0830 h. His finding was similar to the results of Subramanian [15] in Selangor, Malaysia, but differs from that of Yue et al. [16] who identified that *E. kamerunicus* was active between 1130 and 1230 h in China. The differences could be due to differences in the behavior of *E. kamerunicus* or the physiology of the palm trees that is influenced by climatic factors, especially rainfall and temperature [3]. However, these pollinator weevils were less active during the rainy and wet seasons [7,17].

Insect behavior depends on environmental factors, such as temperature and humidity, food sources, the population of the insect itself and the presence of predators. Changes in any of these factors could affect insect behavior. Lemoine et al. [18] reported that temperature influenced the feeding behavior of *Popillia japonica* (Coleoptera: Scarabaeidae) by changing plant interactions towards plants with high levels of nitrogen. Similarly, *Exophthalmus jekelianus* (Coleoptera: Curculionidae) mixes its diet by switching and feeding on different plant species to meet its protein and carbohydrate intake requirements [19]. However, when such an opportunity is restricted, many herbivorous insects adjusted their feeding response to balance the nutrient concentration in their food source [20].

Therefore, the objective of this study was to determine the effect of starvation levels, sexes and sources of *E. kamerunicus* on their diurnal behavior. This study is very important with regard to investigating the pollinating issues in the field, such as lacking male inflorescence, that provides the food sources to *E. kamerunicus* and its ratio that may affect oil palm production. Particularly, this is to find if there are any differences between wild-caught and laboratory-reared adult *E. kamerunicus* behaviors due to different environments. The results of this study are expected to provide new pieces of information and help researchers as well as farmers to manage their plantation, especially concerning the weevil’s population, plantation management and oil palm production.

## 2. Materials and Methods

### 2.1. Sampling Procedure

The samples were collected from seven years oil palm trees on mineral soil at smallholder plantation (2°50′14″ N 101°36′39″ E). Immature oil palm male flower was wrapped by using a muslin sheath to ensure oil palm flowers were free from pollinating weevil and other insects before the experiment. Spikelets of the wrapped male flower were cut at the stages of 75% anthesis or 3rd day of anthesis and brought back to the laboratory for further experiments. Then, on the 2nd day of female flower anthesis, parts of the female flowers were cut off and placed into a 250 mL schott glass bottle to maintain the estragole odor.

Weevils had proven to strive and adapt well in both wet and dry season throughout Malaysia [2]. Therefore, weevil individuals were obtained randomly throughout the experiment. For wild-caught adult *E. kamerunicus*, individuals were obtained by wrapping up dried male spikelets at the post-anthesis stage in the field. Then, adult *E. kamerunicus* were collected and used within two days after emergence [21]. These live adults were left to starve for 24 h before experiments to increase their motivation to search for food sources while unstarved weevils were left for 24 h in another container with a food source.

For laboratory-reared adult *E. kamerunicus*, individuals obtained from the field were reared at the laboratory, with a temperature between 26.5 °C and 28.9 °C on oil palm male inflorescence in a plastic container. Individuals were left to mate and breed within 24 h before being removed from the plastic container. Then, the first progeny was left to develop from egg until adult stage [22] as this generation has higher survival rate to proceed with the experiment. The adults that emerged were used as sample subjects within 48 h after emergence while unstarved weevils were left for 24 h in another container with a food source.

### 2.2. Laboratory Experiment

Two anthesizing oil palm male spikelets and parts of female flowers were tied together vertically and placed into a separate transparent plastic container (12 cm × 11 cm × 10 cm) with a small hole at the bottom for the entrance of *E. kamerunicus* during the experiments, respectively [21,22,23]. A pair of starved *E. kamerunicus* (treatment) were released into the transparent plastic container for each two-hour observation, respectively. Duration and frequency on the diurnal behaviors of *E. kamerunicus* including eating (E), moving (M), resting (R), mating (C), grooming (G) and flying (F) were observed visually and recorded using a mobile phone for two hours between 1130 and 1330 h with natural photoperiod and no artificial lighting involved [12]. For this study, the following behaviors are defined as follows: (i) eating: when the weevil protrudes its snout into the spikelet; (ii) moving: when the weevil is in motion moving from one point to another; (iii) resting: when the weevil is immobile at one point and remains stationary; (iv) mating: when male mounts the back of a female weevil and copulation takes place; (v) grooming: when the weevil uses its legs with the intention of cleaning its body; (vi) flying: when the weevil open its elytra and travels from one point to another using its wing. The experiments were repeated for 10 pairs of *E. kamerunicus* (replicates) for each treatment (sexes, starvation levels and sources of *E. kameruncius*) on both male spikelets and female flowers as food sources.

### 2.3. Data Analyses

A three-way MANOVA was run to determine whether there are significant differences that existed between levels of starvation, sexes and sources of *E. kamerunicus* based on the entire diurnal behavior of *E. kamerunicus*. Wilks lambda was reported to show the variance in the variables contributing to the model, for which values near to zero indicate that most of the variables are important. One-way ANOVA was then performed for each diurnal behavior to determine if it contributed significantly to the levels of starvation, sexes and sources of *E. kamerunicus* differences. Moreover, a paired *t*-test was run to analyze the differences in duration of each diurnal behavior between starved and unstarved *E. kamerunicus*. All analyses were conducted using MINITAB Version 16 software.

## 3. Results

### 3.1. Diurnal Behavior of E. kamerunicus Based on Levels of Starvation, Sexes and Sources of E. kamerunicus on Anthesizing Oil Palm Male Spikelets

Results of the three-way MANOVA on the six diurnal behaviors of *E. kamerunicus* indicated that there were significant differences between levels of starvation (Wilks lambda_6,68_ = 0.38783, *p* = 0.000), sexes (Wilks lambda_6,68_ = 0.75873, *p* = 0.004) and sources of *E. kamerunicus* (Wilks lambda_6,68_ = 0.53827, *p* = 0.000). There was also a significant interaction between levels of starvation and sources of *E. kamerunicus* (Wilks lambda_6,68_ = 0.74246, *p* = 0.002) as well as the interaction between sexes of *E. kamerunicus* and levels of starvation (Wilks lambda_6,68_ = 0.47321, *p* = 0.000) in influencing diurnal behavior of *E. kamerunicus* on oil palm male flowers. However, the interaction between the sources and sexes of *E. kamerunicus* was insignificant (Wilks lambda_6,68_ = 0.84173, *p* = 0.063).

The results of one-way ANOVA on each diurnal behavior of *E. kamerunicus* showed that three diurnal behaviors, M (F_1,78_ = 17.02, *p* = 0.001), R (F_1,78_ = 8.46, *p* = 0.005) and G (F_1,78_ = 27.19, *p* = 0.001) were significantly different between sources of *E. kamerunicus*. There were also significant differences in two diurnal behaviors, F (F_1,78_ = 17.46, *p* = 0.001) and R (F_1,78_ = 42.90, *p* = 0.001), with respect to levels of starvation. However, there were no significant differences in diurnal behaviors of *E. kamerunicus* of different sexes.

Although there was no significant difference for all diurnal behaviors between the sexes of *E. kamerunicus* on oil palm male flowers, the study showed that female *E. kamerunicus* performed better in flying, moving, eating and grooming behavior compared to male *E. kamerunicus*. It was observed that moving behavior recorded the highest frequency (*p* = 0.71, male = 18.75 ± 0.909, female = 18.82 ± 1.59) while the lowest was mating behavior (*p* = 1.00, male = 1.275 ± 0.253, female = 1.275 ± 0.253). However, laboratory-reared *E. kamerunicus* exhibited significantly more flying, resting, moving and grooming behavior compared to the wild-caught *E. kamerunicus* that exhibited more on eating and mating behavior. This study showed that the mean number of moving behaviors recorded the highest frequency (*p* = 0.001, laboratory = 22.2 ± 1.41, wild = 15.375 ± 0.864) while mating behavior recorded the least mean frequency (*p* = 0.81, laboratory-reared = 1.1 ± 0.133, wild-caught = 1.45 ± 0.33). In contrast, starved *E. kamerunicus* exhibited significantly more on flying behavior compared to unstarved *E. kamerunicus* that exhibited more on resting behavior (Figure 1).

### 3.2. Diurnal Behavior of E. kamerunicus as Affected by Levels of Starvation, Sexes and Sources of E. kamerunicus on Oil Palm Female Inflorescence

Results of three-way MANOVA for the six diurnal behaviors of *E. kamerunicus* indicated that there were significant differences between sources (Lambda Wilk_6,68_ = 0.48432, *p* = 0.000), sexes (Wilks lambda_6,68_ = 0.74500, *p* = 0.002) and levels of *E. kamerunicus* (Wilks lambda_6,68_ = 0.44134, *p* = 0.000). There was a significant interaction between levels of starvation and sources of *E. kamerunicus* (Wilks lambda_6,68_ = 0.23253, *p* = 0.000), the interaction between sexes of *E. kamerunicus and* levels of starvation (Wilks lambda_6,68_ = 0.73800, *p* = 0.002) as well as the interaction between the sources and sexes of *E. kamerunicus* (Lambda Wilk_6,68_ = 0.68567, *p* = 0.000) in influencing diurnal behavior of *E. kamerunicus* on oil palm female flower.

Results of one-ANOVA on each diurnal behavior of *E. kamerunicus* showed that two diurnal behaviors, i.e., flying (F_1,78_ = 19.04, *p* = 0.001) and resting (F_1,78_ = 13.13, *p* = 0.001), were significantly different between sources of *E. kamerunicus*. Meanwhile, there were significant differences in two diurnal behaviors, i.e., grooming (F_1,78_ = 12.28, *p* = 0.001) and eating (F_1,78_ = 22.27, *p* = 0.001) between levels of starvation. However, only moving behavior showed significant differences between sexes of *E. kamerunicus* (F_1,78_ = 10.93, *p* = 0.001).

Female *E. kamerunicus* showed significantly more flying, moving, eating and grooming behavior than the male *E. kamerunicus*. The moving behavior had the highest frequency (*p* = 0.001, male = 17.5 ± 1.48, female = 23.15 ± 1.22), while mating behavior had the lowest frequency (*p* = 0.66, male = 0.85 ± 0.127, female = 0.85 ± 0.127). On the other hand, wild-caught *E. kamerunicus* performed significantly more flying, moving and grooming behavior compared to laboratory-reared *E. kamerunicus* that exhibit more resting, eating and mating behaviors. Meanwhile, starved *E. kamerunicus* had higher frequency for flying, resting and eating behaviors compared to fed *E. kamerunicus* that exhibit more moving, grooming and mating behaviors. Results also showed that the mean number of moving behaviors recorded the highest frequency (*p* = 0.44, starved = 19.1 ± 0.99, fed = 21.55 ± 1.74), while mating behavior recorded the least mean frequency (*p* = 0.66, starved = 0.9 ± 0.133, fed = 0.8 ± 0.12).

### 3.3. Duration of Diurnal Behavior for Sources of E. kamerunicus (Wild-Caught and Laboratory-Reared) on Oil Palm Male Spikelets as Affected by Levels of Starvation

Results of the paired t-test showed that there were significant differences in time spent for flying (F) (T = 2.24, dk = 18, *p* < 0.05), moving (M) (T = 7.43, dk = 18, *p* < 0.05), resting (R) (T = 13.88, dk = 18, *p* < 0.05), eating (E) (T = 6.10, dk = 18, *p* < 0.05), grooming (G) (T = 7.04, dk = 18, *p* < 0.05) and mating behavior (C) (T = 6.70, dk = 18, *p* < 0.05) for wild male *E. kamerunicus* among starvation levels. The flying (7.60 ± 1.44), moving (1812 ± 82.6), resting (4028 ± 143) and mating (34 ± 9.92) behavior were performed longer by fed wild-caught male *E. kamerunicus*. In contrast, starved wild-caught male *E. kamerunicus* spent longer eating (2868 ± 126) and grooming (2163 ± 159) compared to other behaviors.

The results of paired t-test showed that there were significant differences in time spent for flying (T = 4.09, dk = 18, *p* < 0.05), eating (T = 13.28, dk = 18, *p* < 0.05), grooming (T = 7.12, dk = 18, *p* < 0.05) and mating (T = 6.07, dk = 18, *p* < 0.05) behavior for wild-caught female *E. kamerunicus* among starvation levels. Meanwhile, the moving (T = 0.62, dk = 18, *p* > 0.05) and resting (T = 1.04, dk = 18, *p* > 0.05) behavior were insignificantly different. The moving (2236 ± 231), resting (2659 ± 121) and mating (253.2 ± 31.2) behavior were longer for unstarved wild-caught female *E. kamerunicus* compared to flying (8.0 ± 0.869), eating (2804 ± 378) and grooming (2097 ± 525) behavior, which were longer for starved wild-caught female *E. kamerunicus*.

However, laboratory-reared male *E. kamerunicus* showed significant differences in time spent for flying (T = 3.36, dk = 18, *p* < 0.05), moving (T = 4.14, dk = 18, *p* < 0.05), resting (T = 9.96, dk = 18, *p* < 0.05), eating (T = 4.13, dk = 18, *p* < 0.05), grooming (T = 3.42, dk = 18, *p* < 0.05) and mating (T = 3.23, dk) = 18, *p* < 0.05) behavior for laboratory-reared male *E. kamerunicus* among starvation levels. Unstarved laboratory-reared male *E. kamerunicus* spent longer flying (7.6 ± 0.884), grooming (2441 ± 159), resting (3070 ± 288) and mating (161.2 ± 30.6). The eating (2720 ± 408) and grooming (2566 ± 530) behaviors, however, were longer for starved laboratory-reared *E. kamerunicus* males.

Meanwhile, there were significant differences in time spent for flying (T = 5.59, dk = 18, *p* < 0.05), moving (T = 6.96, dk = 18, *p* < 0.05), resting (T = 11.64, dk = 18, *p* < 0.05), eating (T = 10.19, dk = 18, *p* < 0.05), grooming (T = 13.41, dk = 18, *p* < 0.05) and mating (T = 3.23, dk = 18, *p* < 0.05) behavior for laboratory-reared female *E. kamerunicus* among starvation levels. Unstarved laboratory-reared female *E. kamerunicus* spent most of the time on moving (2446.8 ± 53.7), resting (2948 ± 185) and mating (161.2 ± 30.6) behaviors. On the other hand, starved laboratory-reared female *E. kamerunicus* spent most of the time flying (23 ± 3.62), eating (3101.6 ± 71.4) and grooming (2295 ± 134).

### 3.4. Duration of Diurnal Behavior of Sources of E. kamerunicus (Wild-Caught and Laboratory-Reared) on Oil Palm Female Flower as Affected by Levels of Starvation

Results of the paired *t*-test showed that there were significant differences in duration per diurnal behavior: flying (T = 4.31, dk = 18, *p* < 0.05), eating (T = 3.25, dk = 18, *p* < 0.05) and grooming (T = 0.88, dk = 18, *p* < 0.05) for wild *E. kamerunicus* males based on levels of starvation. In this study, unstarved wild-caught *E. kamerunicus* had longer flying (13 ± 1.86), resting (4245 ± 204) and mating (25 ± 13.2) behaviors compared to moving (1256 ± 135), eating (3938 ± 457) and grooming (1356 ± 240) behaviors, which were longer for starved wild-caught *E. kamerunicus*.

There were significant differences in the duration of moving (T = 8.04, dk = 18, *p* < 0.05) and eating behaviors (T = 2.84, dk = 18, *p* < 0.05) for wild-caught *E. kamerunicus* females among starvation status. On the other hand, flying (T = 2.01, dk = 18, *p* > 0.05), resting (T = 0.26, dk = 18, *p* > 0.05), grooming (T = 1.64, dk = 18, *p* < 0.05) and mating (T = 0.88, dk = 18, *p* > 0.05) behaviors were insignificantly different. The resting (3771 ± 117) and mating (25 ± 13.2) behaviors were performed longer by unstarved wild-caught *E. kamerunicus* females compared to flying (26.20 ± 8.22), moving (1612 ± 249), eating (2528 ± 101) and grooming (2131.4 ± 95.3) behaviors, which were longer for starved wild-caught *E. kamerunicus* females.

Meanwhile, there were significant differences in time spent for resting (T = 18.47, dk = 18, *p* < 0.05), eating (T = 20.82, dk = 18, *p* < 0.05) and grooming (T = 10.07, dk = 18, *p* < 0.05) behaviors of laboratory-reared *E. kamerunicus* males among starvation status but no significant differences in time spent for flying (T = 0.40, dk = 18, *p* > 0.05), moving (T = 1.79, dk = 18, *p* > 0.05) and mating (T = 1.70, dk = 18, *p* > 0.05) behaviors. The moving (2150 ± 186), resting (3982 ± 149) and mating (13.20 ± 4.12) behaviors were longer for unstarved laboratory-reared *E. kamerunicus* males. On the other hand, starved laboratory-reared *E. kamerunicus* males spent longer flying (3.2 ± 0.854), eating (1119.8 ± 32.3) and grooming (4137 ± 195) compared to other behaviors.

Starvation makes the weevils more active in moving (T = 3.77, dk = 18, *p* < 0.05), resting (T = 2.46, dk = 18, *p* < 0.05), eating (T = 9.03, dk = 18, *p* < 0.05) and grooming (T = 2.90, dk = 18, *p* < 0.05) for laboratory-reared *E. kamerunicus* females. On the other hand, the flying (T = 1.44, dk = 18, *p* > 0.05) and mating (T = 1.70, dk = 18, *p* > 0.05) behaviors were insignificantly different. Starved laboratory-reared *E. kamerunicus* females spent longer moving (2408 ± 187), resting (3540 ± 349), grooming (921.2 ± 63.6) and mating (13.20 ± 2.82) compared to other behaviors.

## 4. Discussion

A study on the diurnal behavior of the oil palm pollinator, *E. kamerunicus,* is very important to understand problems related to pollination activities. In this study, there were significant differences in the frequency of diurnal behaviors of *E. kamerunicus,* as affected by levels of starvation, sexes and sources of *E. kamerunicus* on both oil palm male spikelets and female flowers. At the same time, our results demonstrate that one-way ANOVA showed no significant differences for each diurnal behavior between sexes of *E. kamerunicus* on oil palm male spikelets. This may be due to the *E. kamerunicus* itself that is a host-specific insect pollinator, in which it needs the male spikelets to serve as a food source as well as a breeding site [6,24,25].

Female *E. kamerunicus* were more active and spent a long time on feeding (E) and moving (M) compared to males, even though the differences were insignificant. This is probably because they have to focus more on searching for food to gain energy as well as to search for a breeding site by using their long snout. Similarly, female *Rhynchophorus ferrugineus* (Coleoptera: Curculionidae) also showed this trend, where it actively moves in a search for food sources, as diet intake significantly affects female oviposition and the egg-laying process [12,26]. In contrast, male *E. kamerunicus* were more active in resting after the mating process, as they may not need as much energy as the female *E. kamerunicus* for the egg-laying process.

Additionally, female *E. kamerunicus* were also active in grooming behavior (G). As explained by Mc Iver [27], the male *E. kamerunicus* groom their feet to enhance the ability to grasp the female during the mating process as well as to improve their sensory abilities for detecting food sources by using chemoreception and mechanoreception sensilla. Jacquet et al. [28] reported that grooming behavior plays an important role in mating and calling behavior. Zhang et al. [13] also stated that grooming behavior enables female *Eucryptorrhynchus brandti* (Coleoptera: Curculionidae) to search their breeding site effectively. They also point out that grooming behavior is really important and can be considered as a positive biological indicator, in which the more the weevils groom, the healthier they are, having clean gustatory and olfactory areas and thermosensilla [13].

Laboratory-reared insects exhibit changes in physical traits as well as behavior as compared to wild ones [29]. In this study, different developments were demonstrated by the laboratory-reared *E. kamerunicus,* in which they have a smaller body size than the wild-caught *E. kamerunicus*. This showed the flying and eating behavior of lab-reared weevils was higher compared to wild-caught weevils (Figure 1). This could be due to different diets, spaces and environments during the larvae and adult stages. This contention is supported by Manley et al. [30], who studied *Oryctes rhinoceros* (Coleoptera: Scarabaeidae). They reported that laboratory-reared *O. rhinoceros* has a significant effect on biology itself, in which wild *O. rhinoceros* showed higher reproductive ability and had a larger body size compared to laboratory-reared *O. rhinoceros.*

Starvation influenced the diurnal behavior of *E. kamerunicus* on both oil palm flowers in which starved weevils spend more in feeding behavior compared to unstarved weevils (Figure 1). The unstarved weevils have to groom for much longer than starved weevils on both oil palm flowers [28]. In addition, starvation leads weevils to respond to physiological needs. This assumption is supported by the significant differences found between unstarved and starved weevils in the total duration of grooming behavior. Unstarved *E. kamerunicus* also showed more frequent mating behavior compared to starved *E. kamerunicus*. This could be due to starved *E. kamerunicus* being busy searching for food rather than finding a mate. Starving for 24 h increased their motivation to search for food and reduced the potential of *E. kameruncius* to find a mate. Oku et al. [31] concluded that starvation of *Phyllotreta nemorum* (Coleoptera: Chrysomelidae) stimulates and increase their movement activity in search of a food source.

Starvation in females may affect their reproductive ability, as reported by Barry [32] and Barry [33], who stated that starving *Pseudomantis albofimbriata* (Mantodea: Mantidae) females negatively affected their reproductive ability and interest towards males for mating. Food intake, size and condition of the body also influence the production of pheromones and attractiveness to the mate [34]. Mating behavior begins with the male *E. kamerunicus* touching the final back parts of the female body with its protarsus and antennae. At this moment, some chemical recognition may have occurred between both. Then, the male grabs the female with the legs and positions the posterior part of its body next to the female pygidium and inserts the aedeagus to begin copulation. This is similar to the observation reported by Ferreira, Gomes and Rodrigues [35] on the mating behavior of *Leucothyreus albopilosus* (Coleoptera: Scarabaeidae). On the other hand, starved male *E. kamerunicus* still performed mating behavior (K) but spent a shorter time compared to unstarved male *E. kamerunicus*. This condition will reduce the tandem duration that may affect the reproductive success of the male insect [1].

Flying behavior (F) of starved *E. kamerunicus* was higher compared to unstarved *E. kamerunicus* (Figure 1). This showed that starvation stimulates flying (F) activity in searching for food sources. Flying behavior was performed to find a mate for the mating process but was interrupted by a starving condition that focused on finding food rather than finding a mate. Similar results were demonstrated by a study of Oku et al. [31] on *Phyllotreta nemorum* (Coleoptera: Chrysomelidae) in Japan. Starved *P. nemorum* stimulated flight activity (male: 43.3%; female: 33.3%) more than unstarved *P. nemorum* (male: 19.4%; female: 6.3%). Therefore, *E. kamerunicus* could find their mate only if they had a sufficient food source.

Moving behavior (M) of starved male *E. kamerunicus* was longer. On the other hand, unstarved female *E. kamerunicus* spent a long time on moving behavior (Figure 1). This is probably due to the availability of food sources on oil palm male flowers [12]. The starved *E. kamerunicus* requires higher movement to search for food on oil palm male flowers rather than on oil palm female flowers. Anggraeni et al. [36] reported that there are three compounds in oil palm male flowers, namely palmitic acid, estragole and 1-dodecyne, while there are four compounds in oil palm female flowers, namely chloroacetic acid, 4-tetra decyl ester, palmitic acid, farnesol and squalene [11]. The farnesol and squalene compounds are only available in oil palm female flowers, and this may be the cause of *E. kamerunicus* visiting the oil palm female flower for a short period. By observation, there was a feeding behavior in starved *E. kamerunicus* on oil palm female flowers but not as extreme as the feeding behavior on oil palm male flowers (Figure 1). The *E. kamerunicus* was observed inserting its rostrum into the oil palm female flower, most probably searching for nectar [13].

## 5. Conclusions

In conclusion, starvation prevented the weevils from other activities because they had to focus more on searching for food to gain energy before performing other behaviors, which are very closely related to each other and, therefore, fed weevils were more active in mating and resting compared to unfed weevils. Female weevils tend to move more actively as they need to perform the egg-laying process. However, both male and female weevils are involved in oil palm pollination as they are attracted to the anise-like odor of estragole that is released by male and female oil palm flowers. This finding could provide information for oil palm production if sustainable management of an oil palm plantation is carried out effectively to maintain the numbers of oil palm male flowers, as they can provide the breeding site for *E. kamerunicus.*

## Figures and Tables

**Figure 1 insects-13-00940-f001:**
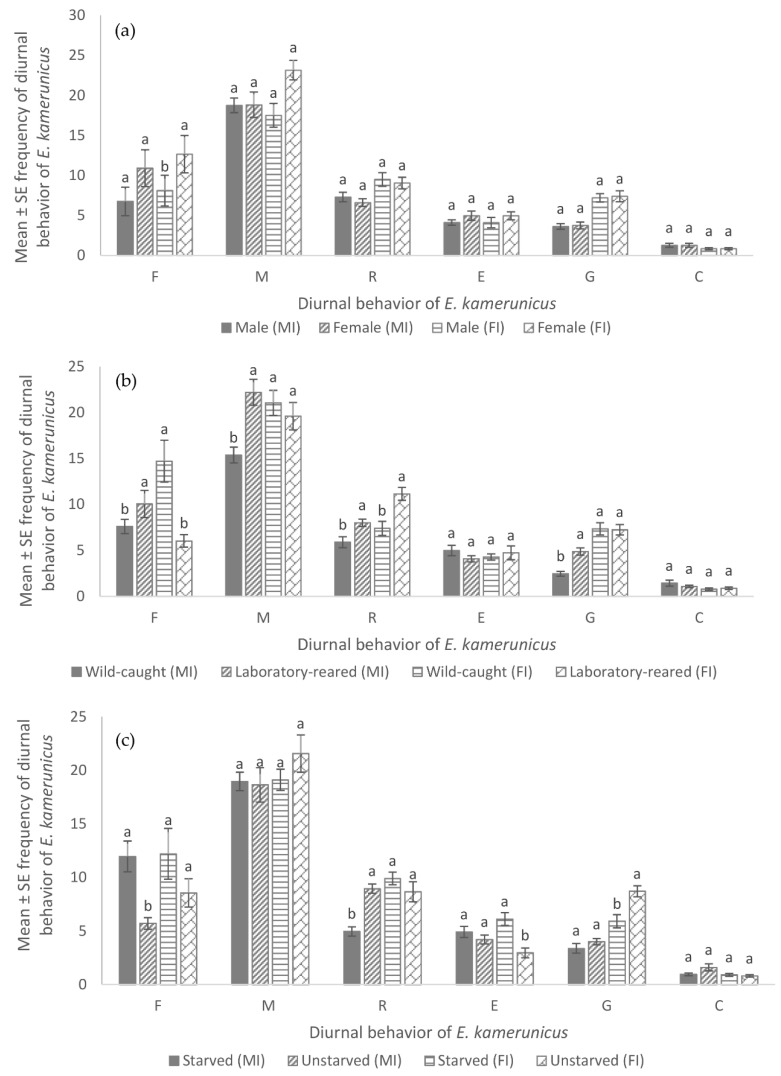
Mean (± SE) frequency of diurnal behavior of *E. kamerunicus* between (**a**) sexes (**b**) sources of *E. kamerunicus* and (**c**) levels of starvation on oil palm male inflorescence (MI) and female inflorescence (FI); F = flying, M = moving, R = resting, E = eating, G = grooming, C = mating/copulate. Note: same case letters indicate that there is no significant difference (*p* < 0.05).

## Data Availability

Data are contained within the article.

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
