# Peer review of "Starvation Levels Affect Behaviors of Wild-Caught and Laboratory-Reared Oil Palm Pollinator Weevil, Elaeidobius kamerunicus (Coleoptera: Curculionidae)"

_insects, 2022, doi:10.3390/insects13100940_

Round 1

Reviewer 1 Report

The manuscript foucsed on starvation levels and sexes affect diurnal behavior of Elaeidobius kamerunicus, This study are expected to provide new pieces of information and help researchers as wellas farmers to manage their plantation expecially concerning the weevil’s population, plantation management, and oil palm production. However, this manuscript need improved as following.

Line 4, Elaeidobius Kamerunicus, K letter do not capital letter.

Line33, Keywords: Coleoptera; Curculionidae, changed Elaeidobius kamerunicus

Line174-192, Figure.2 showed wrong letters, a high number should be showed “a”, a low number should be showed “b”in the same Figure, but this Figure.2 showed opposited letters.

Line 242-249, Figure.3 showed the same wrong in Figure.2

Line287-324, Table 5, Table 6 showed the same wrong in Figure.3 and Figure.2, a high number should be showed “a”letter, a low number should be showed “b” in the same Figure or Table.

Reviewer 2 Report

This is an interesting study on a major plam pollinator weevil exotic to Malaysian palm production. Oil palm cultivation occurs due to human formal agricultural practices. Therefore, the title of the study is misleading. This need to be modified for example Foraging Behavior of Oil Palm Pollinator Weevil (Coleoptera: Curculionidae) in the Laboratory and Open Field Conditions in Malaysia

Provide a statement in the Introduction who are the beneficiary of this weevil for example oil palm producers, industry, ...If we know a figure of economic impact per year. This will help to the readers. 

The manuscript needs proper formatting based on the Insects author instructions: https://www.mdpi.com/journal/insects/instructions.

Reviewer 3 Report

 The authors conducted a series of very simple experiment. A large set of raw data was obtained and correctly analyzed. The main problem of the study is that based on this analysis the authors are able to make only two rather trivial conclusions. First (as, of course, should be expected) “starvation prevented the weevils from other activities because they have to focus more on searching for food” (lines 421-422). Second, which is also common in insects “female weevils tend to move more actively... to perform egg laying” (lines 423-424).  Although even this information can be important for palm growing, I doubt that such a large dataset (the Results section takes 10 pages with 3 figures and 6 tables) is really necessary to make these two simple conclusions. Therefore I strongly recommend substantial (at least 50%) reducing the manuscript. Below I suggest some ways to do it.  

1) For example, MANOVA results are not really important for the study because (Table 2 + Fig. 2) and (Table 4 + Fig. 3) show the significance and sign of the influence of the 3 experimental factors on the frequency of the 6 studied types of behavior on male and female flowers, correspondingly. Possibly, Tables 1 and 3 can be omitted? Moreover, considering that the data on the frequency of different types of behavior in different experimental treatments and significances of the differences between treatments are clearly presented in the figures, it is also possible to omit also tables 2 and 4 with one-way ANOVA results.

2) Another way to reduce the manuscript: in the Results each type of behavior is described with two parameters: duration and frequency. In the Discussion, both parameters are mixed up. Moreover, in the Conclusions only types of behavior are mentioned (without the separation of their duration and their frequency). Possibly, only data for duration (as more practically important characteristic) can be given? Otherwise, possibly, duration and frequency can be combined and converted into some integrative parameter? 

3) One more suggestion: figures 2 and 3 give the same data for male and female palm inflorescence. Thus, it is reasonable to combine these figures by using not 2 but 4 types of bar filling. This would make the paper shorter and, in addition, would allow easy visual comparison of the data for male and female inflorescence.

Possibly, the authors will find another ways. In any case, the manuscript should be much shorter: I guess, the results can be presented on 3-4 pages with 1-2 figures or tables.

Finally, I think that it would not be reasonable to make any minor comments to the present version of the manuscript because it has to be almost completely rewritten before publication. However, some (preliminary) minor comments are listed below.

Lines 2-4: Please, check and correct capital letters in the title. All nouns, as well as insects order and family should start with capital letter, whereas species should not.

Lines 37-38 Latin species and genera names should be in italics here and everywhere in the text.

Line 112: As far as I understand, experiments were also conducted at this “room temperature”, not under controlled conditions? If yes, what about photoperiod? Was it natural? If yes, what was the sunrise and sunset time? If you have some artificial lighting in the laboratory, when it was turned on and off? Without this information, the indication of the observation period (line 127) is just senseless: not “clock time” but time in relation to the real light-dark rhythm is important.   

Line 123: It seems that Figure 1 was lost.

Lines 124-125. Please, clearly explain how these parameters were calculated. Duration (as I can guess from Tables 5 and 6) means the total duration for the 2 h observation period, not the duration of each particular case of the given behavior? 

Line 423: Why “On the other hand”? There are no contradictions between the first and second sentences of the Conclusions: starved beetles are focused on searching for food and therefore fed beetles are more active on other types of behavior, including mating.

Reviewer 4 Report

Dear authors,

My comments are in the attached file. I recommend that you revise the manuscript according to the rules of the journal, and also consider my comments.

Round 2

Reviewer 2 Report

After the revision, the manuscript reads well and has improved. 

Reviewer 3 Report

Line 175: replace “the study was observed that” with “the study showed that” or ‘we observed that”.

Line 217: I guess, you mean “no significant difference between insect males and females”, not between male and female inflorescence? Anyway, this should be clearly stated in the legend.   

Line 288: species name should not start with capital letter.

Reviewer 4 Report

The authors took into account my comments on the manuscript and answered on questions. Accept in present form.

Author Response

No comments from this reviewer